# A Comparison of Ensemble and Dimensionality Reduction DEA Models Based on Entropy Criterion

## Parag C. Pendharkar

Information Systems School of Business Administration, Pennsylvania State University, Harrisburg 777 West Harrisburg Pike, Middletown, PA 17057, USA; pxp19@psu.edu; Tel.: +1-717-948-6028

**Abstract:** Dimensionality reduction research in data envelopment analysis (DEA) has focused on subjective approaches to reduce dimensionality. Such approaches are less useful or attractive in practice because a subjective selection of variables introduces bias. A competing unbiased approach would be to use ensemble DEA scores. This paper illustrates that in addition to unbiased evaluations, the ensemble DEA scores result in unique rankings that have high entropy. Under restrictive assumptions, it is also shown that the ensemble DEA scores are normally distributed. Ensemble models do not require any new modifications to existing DEA objective functions or constraints, and when ensemble scores are normally distributed, returns-to-scale hypothesis testing can be carried out using traditional parametric statistical techniques.

**Keywords:** data envelopment analysis; dimensionality reduction; ensembles; exhaustive state space search; entropy

## 1. Introduction

Data envelopment analysis (DEA) is a prominent technique for the non-parametric relative efficiency analysis of a set of decision-making units (DMUs) drawn from a similar production process [1]. DEA models are used in both operation research and data mining literature [2]. Some of the traditional properties of production functions, such as the monotonicity and convexity of the inputs and outputs, that are fundamental in DEA models are often found to be attractive in some data mining models where datasets are noisy and model resistance to learning noise is necessary [3]. An important aspect of DEA models is the reliability of DMU efficiency scores. It is generally accepted that the DEA efficiency estimates are reliable when the sample size is large [4]. Since the reliability of the DEA scores is dependent on the sample size, Cooper et al. [5] have suggested the following rule for the minimum number ($n$) of DMUs for reliable DEA analysis (each DMU has $m$ inputs and $s$ outputs):

$$n \geq max\{3(m+s), \, m \times s\} \tag{1a}$$

For small-size datasets, where violations of the minimum number of DMUs specified by Equation (1a) frequently occur, dimensionality reduction (also known as variable reduction or variable selection) approaches are frequently used to select a subset of variables to satisfy Equation (1a). A variety of variable selection approaches are available in the literature. Among these variable selection approaches are statistical [6], regression [7], efficiency contribution measure [8], bootstrapping [9], hypothesis testing [10], variable aggregation [11] and statistical experiment designs [12]. Variable selection approaches are criticized extensively for applying parametric procedures and linear relationship assumptions for selecting variables to determine an unknown non-linear and non-parametric efficiency frontier. Nataraja and Johnson [13] provide a good description of some of these procedures and their pros and cons.

Pendharkar [14] proposed a competing approach to the dimensionality reduction/variable selection problem called the ensemble DEA. In his approach, traditional DEA analysis is conducted for all possible input and output combinations, and the efficiency scores of each DEA model for each DMU are averaged as an ensemble efficiency score for a DMU. Drawing from machine learning literature, Pendharkar [14] showed that the ensemble efficiency score is a reliable estimate of the "true" efficiency of a DMU. Even for small datasets, certain combinations of inputs will satisfy the criterion set by Equation (1a), while others will violate it, but the average ensemble score will be closer to the true efficiency of the DMU and will be reliable. Pendharkar [14] also proposed an exhaustive search procedure to generate all possible input and output combinations, and proposed a formula to compute the number of unique DEA models that need to be run to compute an average ensemble score. This number $N$ of unique DEA models may be computed using the following formula:

$$N = \left(\sum_{i=1}^{m} \binom{m}{i}\right) \times \left(\sum_{i=1}^{s} \binom{s}{i}\right) = (2^m - 1) \times (2^s - 1). \tag{1b}$$

Using Banker et al.'s [15] variable-returns-to-scale (VRS) DEA BCC model, and data and models obtained from a few studies in the literature, Pendharkar [14] showed that the ensemble DEA model provides a better ranking of DMUs than the models proposed in a few studies from the literature.

This research investigates the additional properties and statistical distribution of the ensemble DEA model scores. It is shown that there are added benefits of ensemble efficiency scores. In particular, the ensemble efficiency scores maximize entropy, meaning that the DMU ranking distribution generated by the ensemble efficiency scores has a lower bias when compared to some competing radial and non-radial variable selection models recently reported in the literature, and second, the ensemble efficiency scores may be normally distributed under certain restrictive assumptions. The normal distribution of the efficiency score feature is particularly attractive because returns-to-scale hypothesis testing may be conducted by using traditional difference-in-means parametric statistical procedures. Both of these features are tested using data and models reported in a published study [16]. The rest of the paper is organized as follows: In Section 2, the basic DEA radial and non-radial models, ensemble DEA model and Entropy criterion for comparing different DEA models are described. In Section 3, using Iranian gas company data, the results of ensemble DEA models are compared with the results of variable selection models used in Toloo and Babaee's [16] study. Additionally, in Section 3, the properties of the ensemble DEA scores are investigated in terms of the entropy criterion and their statistical distributions. In Section 4, the paper concludes with a summary and directions for future research.

## 2. DEA Preliminaries, Ensemble DEA Model, Entropy Criterion for DEA Model Comparisons and Statistical Distribution of Ensemble Scores

The basic DEA model assumes $n$ DMUs, with each DMU consisting of $m$ different inputs that produce $s$ different outputs. The input and output vectors are semi-positive, and for DMU$_j$ ($j = 1, \ldots, n$), the space for the input and output vectors $(x_j, y_j) \in \mathbb{R}_+^{m+s}$. For a DMU$_o$, its relative efficiency may be computed by using the linear programming model under the constant returns-to-scale assumption. This efficiency is computed by solving the following model:

$$\max \sum_{r=1}^{s} u_r y_{ro}, \tag{2a}$$

subject to:

$$\sum_{i=1}^{m} v_i x_{io} = 1 \tag{2b}$$

$$\sum_{r=1}^{s} u_r y_{rj} - \sum_{i=1}^{m} v_i x_{ij} \leq 0 \quad for\ all\ j = 1, \ldots, n \tag{2c}$$

$$v_i, u_r \geq \varepsilon \ \text{for all } i = 1, \ldots, m \text{ and } r = 1, \ldots, s \tag{2d}$$

where $v_i$ and $u_r$ are the weights associated with the $i$th input and $j$th output, respectively. The constant $\varepsilon > 0$ is infinitesimally non-Archimedean. The model (2a)–(2d) is often called the primary CCR model [1], and its dual is written as follows:

$$minimize \ \theta - \varepsilon \left( \sum_{i=1}^{m} s_i^- + \sum_{r=1}^{s} s_r^+ \right), \ \ldots\ldots \tag{2e}$$

subject to:

$$\sum_{j=1}^{n} \lambda_j x_{ij} + s_i^- = \theta x_{io}, \quad i = 1, \ldots, m \tag{2f}$$

$$\sum_{j=1}^{n} \lambda_j y_{rj} - s_r^+ = y_{ro}, \quad r = 1, \ldots, s, \text{ and} \tag{2g}$$

$$\lambda_j, s_i^-, s_r^+ \geq 0 \ \text{for all } i = 1, \ldots, m; j = 1, \ldots, n; r = 1, \ldots, s \tag{2h}$$

The VRS BCC model augments the system (2e)–(2h) by adding the following constraint:

$$\sum_{j=1}^{n} \lambda_j = 1$$

The aforementioned models are radial DEA models that are criticized for not providing input or output projections (for inefficient DMUs) that satisfy Pareto optimality conditions [17]. Fare and Lovell [18] independently proposed radial DEA models that allow for input or output reductions at variable rates. The radial version of the CCR model is mathematically represented in the following dual form:

$$minimize \ \frac{1}{m} \sum_{i=1}^{m} \theta_i$$

subject to:

$$\sum_{j=1}^{n} \lambda_j x_{ij} \leq \theta_i x_{io}, \quad i = 1, \ldots, m$$

$$\sum_{j=1}^{n} \lambda_j y_{rj} \geq y_{ro}, \quad r = 1, \ldots, s$$

$$\theta_i \leq 1, \quad i = 1, \ldots, m \ \ldots\ldots$$

$$\lambda_i \geq 0, \quad j = 1, \ldots, n$$

Pendharkar [14] proposed an ensemble DEA model based on the popularity of ensemble models in machine learning literature. The ensemble DEA model requires an exhaustive search procedure using a binary vector $z$ whose components indicate whether an input or output is considered in performing DEA analysis. The dimension of this binary vector is ($m + s$). Figure 1 illustrates the $z$ vector and exhaustive search tree for two-input-and-one-output datasets. The exhaustive tree is pruned (dotted edges) for models that have either no inputs or no outputs. DEA analysis is then conducted on the remaining models, and the efficiency results of each model for each DMU are averaged and used as ensemble DEA scores. To illustrate the ensemble DEA approach on a two-input-and-one-output dataset, a CCR DEA analysis using partial Cobb–Douglas production function data on US economic growth between 1899 and 1910 [19] is conducted. Table 1 illustrates the results of our DEA analysis and resulting ensemble scores. The two inputs were labor in person-hours worked per year and the amount of capital invested. The output was the total annual production. The results of the analysis show that the traditional DEA with $z$ = [111] does not provide unique rankings (for the years 1901

and 1902 receive the same efficiency score), but the ensemble DEA model provides unique DMU rankings. Pendharkar's [14] study provides a theoretical basis for the reliability of ensemble DEA scores.

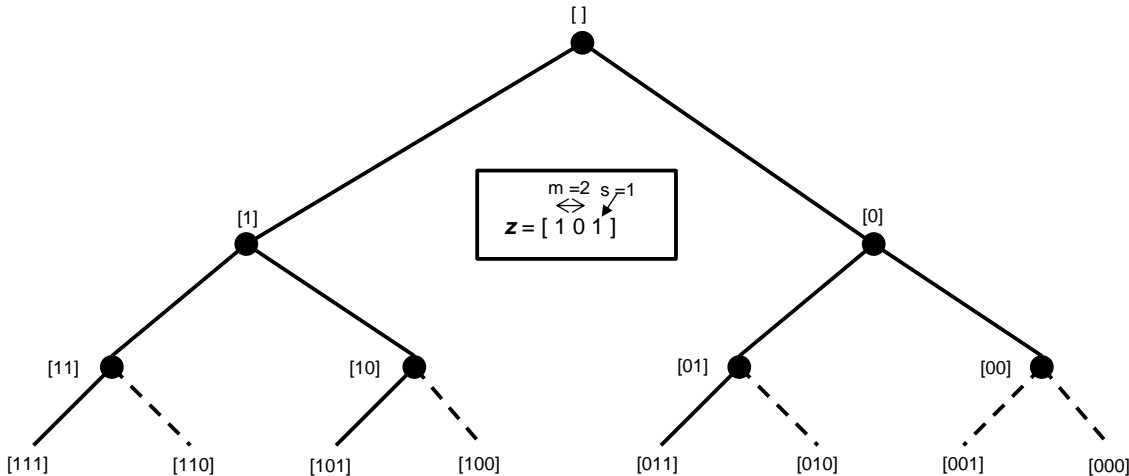

**Figure 1.** Exhaustive Search Tree for possible unique combinations of two-input-one-output datasets.

**Table 1.** Ensemble data envelopment analysis (DEA) scores for 1899–1910 US economic growth data.

| Year | Production | Labor | Capital | DEA Model Efficiencies | | | Ensemble Score |
| --- | --- | --- | --- | --- | --- | --- | --- |
| | | | | $z$ = [111] | $z$ = [101] | $z$ = [011] | |
| 1899 | 100 | 100 | 100 | 0.681 | 0.681 | 0.665 | 0.676 |
| 1900 | 101 | 105 | 107 | 0.722 | 0.722 | 0.678 | 0.707 |
| 1901 | 112 | 110 | 114 | 0.693 | 0.693 | 0.689 | 0.692 |
| 1902 | 122 | 117 | 122 | 0.693 | 0.681 | 0.693 | 0.689 |
| 1903 | 124 | 122 | 131 | 0.720 | 0.720 | 0.714 | 0.718 |
| 1904 | 122 | 121 | 138 | 0.770 | 0.770 | 0.758 | 0.766 |
| 1905 | 143 | 125 | 149 | 0.793 | 0.710 | 0.793 | 0.765 |
| 1906 | 152 | 134 | 163 | 0.809 | 0.730 | 0.809 | 0.783 |
| 1907 | 151 | 140 | 176 | 0.836 | 0.794 | 0.836 | 0.822 |
| 1908 | 126 | 123 | 185 | 1.000 | 1.000 | 1.000 | 1.000 |
| 1909 | 155 | 143 | 198 | 0.921 | 0.870 | 0.921 | 0.904 |
| 1910 | 159 | 147 | 208 | 0.941 | 0.891 | 0.941 | 0.924 |

The maximum entropy (ME) principle has been applied to DEA DMU ranking distribution [20] and model comparisons [21]. The ME principle measures the DMU ranking bias by using a more general family of distributions [22]. Several statistical distributions can be characterized as ME densities [23]. The ME distributions are the least biased distributions obtained by imposing moment constraints that are inherent in the data [21]. To obtain the ME for a given set of DMUs and their efficiencies, normalized ranks are first obtained by computing $\frac{\theta_i^*}{\sum_{i=1}^n \theta_i^*}$, for each DMU, and then computing the ME for a certain model $z$ as follows:

$$\text{ME}^z = - \sum_{i=1}^n \left( \frac{\theta_i^*}{\sum_{i=1}^n \theta_i^*} \right) ln \left( \frac{\theta_i^*}{\sum_{i=1}^n \theta_i^*} \right)$$

The ME for the DEA models in Table 1 are $\text{ME}^{111}$ = 2.4768, $\text{ME}^{101}$ = 2.4775 and $\text{ME}^{011}$ = 2.4757. The model with labor as an input and production as an output ($z$ = [101]) has the highest entropy and has the least bias, with a maximum difference between DMU efficiencies for closely ranked DMUs for the years 1901 and 1902. The ensemble entropy is 2.4769, and since it is an average of all $z$-vector combinations, the comparison benchmark for ensemble entropy is the model with $z$ = [111].

The ensemble entropy is higher than the benchmark. The highest possible entropy value or upper bound (UB) for a model is given by the following expression:

$$\text{ME}^{\text{UB}} = -n \times \left( \left( \frac{1}{n} \right) ln \left( \frac{1}{n} \right) \right) \tag{2i}$$

The $\text{ME}^{\text{UB}}$ for the data in Table 1 is 2.485, and the ensemble entropy is very close to the maximum value. It is important to note that obtaining the maximum value is not always desirable, but it provides a theoretical benchmark estimate for a completely unbiased normalized DMU score distribution.

To compute ensemble efficiency scores, an $n \times m$ matrix $E$ of DEA efficiency scores is necessary. The rows of such a matrix are the numbers of DMUs, and the columns are the numbers of models given by the numbers of eligible models considered in computing ensemble efficiency scores. This number of eligible models will have an upper bound given by $N$, computed using Equation (1b). The elements of this matrix will be efficiency scores for each DMU computed for a given model identified by column number. Figure 2 illustrates a five-DMU-and-five-model matrix. The ensemble efficiency score ($\theta_i^E$) for each DMU is computed using the following formula:

$$\theta_i^E = \frac{\sum_{j=1}^m \theta_{ij}^*}{m} \tag{2j}$$

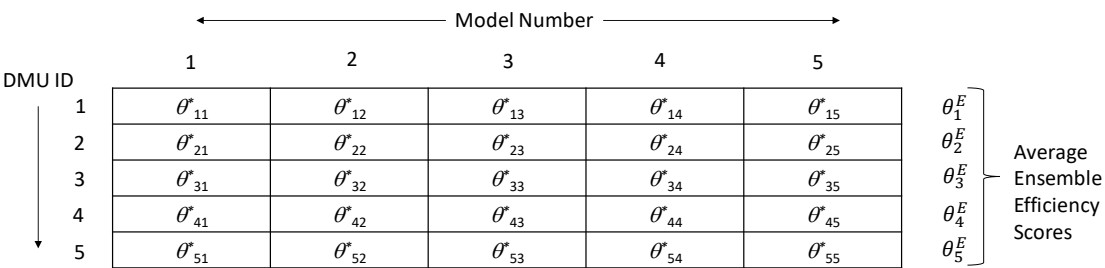

**Figure 2.** An illustration of $5 \times 5$ ensemble efficiency score matrix.

A few observations can be made about any row $i \in \{1, \dots, n\}$ of the ensemble efficiency score matrix. First, all the elements of a given row are an independent computation of efficiency scores by the same DMU under a different model number with its unique set of input(s) and output(s). Second, in all the elements of a given row, the DMU is maximizing its efficiency given its model constraints. Thus, each row represents independent evaluations by a DMU under the maximum decisional efficiency (MDE) principle [24]. The MDE principle was introduced by Troutt [25] to develop a function to the aggregate the performance of multiple decision-makers. The underlying assumption of the MDE principle is that all decision-makers seek to maximize their decisional efficiencies. Troutt [26] later used the MDE approach to rank DMUs and showed that DMUs deemed efficient under MDE are also efficient when ranked using the DEA. For a linear aggregator function, such as the one used in Equation (2j), Troutt [26] illustrated that the decisional efficiencies $\theta$ can be described by the following probability density function (pdf):

$$g(\theta) = c_\alpha e^{\alpha\theta}, \ \alpha > 0 \ \text{and} \ \theta \in [0, 1] \tag{2k}$$

The pdf in (2k) is monotone, increasing on its interval with a mode at $\theta = 1$ (see Figure 5 for illustration). Using the laws of probability, the value of $c_\alpha = \alpha \left( e^\alpha - 1 \right)^{-1}$. Since each element in a given row of the ensemble efficiency score matrix is an independent evaluation by a decision-maker

(i.e., a DMU in an ensemble model) trying to maximize its decisional efficiency $\theta_{ij}^*$ for $j = \{1, \ldots, m\}$, the probability density function for each row (DMU) can be written as:

$$g(\theta_i) = c_{\alpha_i} e^{\alpha_i \theta_i}, \; \alpha_i > 0 \text{ and } \theta_i \, \epsilon \, [0, 1] \tag{2l}$$

The central limit theorem mentions that the cumulative distribution functions (cdfs) of the sums of independently identically distributed random variables asymptotically converge to a Gaussian cdf. The ensemble efficiency scores are normalized sums of independent efficiency assessments that will be distributed with a pdf given by (2l). These sums can be considered independent and identically distributed if $\alpha_1 = \alpha_2 = \ldots = \alpha_n$. Under the restrictive assumption that $\alpha_1 = \alpha_2 = \ldots = \alpha_n$, the ensemble efficiency scores are guaranteed to asymptotically converge to a normal distribution by the central limit theorem. In practice, however, the ensemble efficiency scores are not entirely random or perfectly identically distributed (due to the slight likely variation of Equation (2l)'s $\alpha_i$ parameters for each row), and each ensemble model does introduce a degree of mild randomization. For mild differences in the row pdf parameters $\alpha_i$, where $\alpha_1 \approx \alpha_2 \approx \ldots \approx \alpha_n$, the ensemble efficiency scores are likely to be normally distributed. A reader may note that under ideal conditions, where $\alpha_1 = \alpha_2 = \ldots = \alpha_n$ and individual DMU scores follow Equation (2l)'s distribution, the entropy of the ensemble scores will be highest and close to the highest upper bound mentioned in Equation (2i) because the distribution in Equation (2i) has a mode of 1 (see Figure 5). Thus, it may be argued that the likelihood of normality of the ensemble scores increases when the entropy of the ensemble scores is closer to its upper bound given by Equation (2i). It is important to note that an entropy equal to the exact value of the upper bound given by Equation (2i) is undesirable because at that value, the distribution is a uniform distribution where all the DMUs are fully efficient for all the models. The entropy of the pdf in Equation (2k) is maximized on the interval [0, 1] when the mean of the distribution is greater than 0.5 [27]. Additionally, another important aspect of the distribution of the ensemble efficiency scores is that both the rows and columns of ensemble efficiency scores (Figure 2) play a role in the pdf of the ensemble efficiency scores because the rows represent sampling from the MDE distributions and the columns represent sampling from the distribution of the sums of independent variables. Larger sample sizes increase the statistical reliability and robustness of the results.

## 3. Comparing Variable Selection Models and Ensemble Model Using Gas Company Data and Entropy Criterion

For small datasets, many input or output variables are aggregated so that the selected variables satisfy the heuristic given in Equation (1a). There are two problems with all the variable selection approaches. First, they use an artificial criterion to select variables for a non-linear and non-parametric approach. Any artificial/subjective criterion will make some assumptions that are harder to justify. Second, these techniques have several selection parameters and thresholds that often lead to inconsistencies in applying these techniques. For example, Toloo and Babaee [16] illustrate three problems with a variable selection approach and suggested an improved approach. By contrast, the ensemble DEA approach does not make any assumptions, and for small datasets, trying out different input and output combinations and aggregating efficiency scores provide more reliable efficiency estimates than variable selection models. Part of the reason for the stability of ensemble DEA efficiency scores is that, even for small datasets, some DEA models in an ensemble will always satisfy the heuristic given in Equation (1a), which will increase the reliability of the ensemble efficiency scores due to model averaging. This stability of ensemble efficiency scores is illustrated by comparing ensemble scores with the results of models from Toloo and Babaee's [16] study and using the entropy criterion.

To compare the results, the dataset from Toloo and Babaee's [16] study is used. The dataset consists of three inputs and four outputs from an Iranian gas company. The inputs are budget ($x_1$), staff ($x_2$) and cost ($x_3$). The outputs are customers ($y_1$), the length of the gas network ($y_2$), the volume delivered ($y_3$) and gas sales ($y_4$). Table 2 lists these data. Table 3 lists the efficiency scores of the ensemble DEA

with the CCR and BCC models and models used by Toloo and Babaee [16]. Using formula (1b), a total of 105 unique DEA models were used to compute the DEA ensemble efficiency score.

**Table 2.** The Iranian gas company dataset.

| DMU | $x_1$ | $x_2$ | $x_3$ | $y_1$ | $y_2$ | $y_3$ | $y_4$ |
|---|---|---|---|---|---|---|---|
| 1 | 177,430 | 401 | 528,325 | 801 | 41,675 | 77,564 | 201,529 |
| 2 | 221,338 | 1094 | 1,186,905 | 803 | 34,960 | 44,136 | 840,446 |
| 3 | 267,806 | 1079 | 1,323,325 | 251 | 24,461 | 27,690 | 832,616 |
| 4 | 160,912 | 444 | 648,685 | 816 | 23,744 | 45,882 | 251,770 |
| 5 | 177,214 | 801 | 909,539 | 654 | 36,409 | 72,676 | 443,507 |
| 6 | 146,325 | 686 | 545,115 | 177 | 18,000 | 19,839 | 341,585 |
| 7 | 195,138 | 687 | 790,348 | 695 | 31,221 | 40,154 | 233,822 |
| 8 | 108,146 | 152 | 236,722 | 606 | 23,889 | 37,770 | 118,943 |
| 9 | 165,663 | 494 | 523,899 | 652 | 25,163 | 28,402 | 179,315 |
| 10 | 195,728 | 503 | 428,566 | 959 | 43,440 | 63,701 | 195,303 |
| 11 | 87,050 | 343 | 298,696 | 221 | 9689 | 17,334 | 106,037 |
| 12 | 124,313 | 129 | 198,598 | 565 | 21,032 | 30,242 | 61,836 |
| 13 | 67,545 | 117 | 131,649 | 152 | 10,398 | 14,139 | 46,233 |
| 14 | 47,208 | 165 | 228,730 | 211 | 9391 | 13,505 | 42,094 |

**Table 3.** The results of experiments.

| DMU | Ensemble CCR | Ensemble BCC | Non-Radial [&] | Radial [&] |
|---|---|---|---|---|
| 1 | 0.87 (0.15) | 0.95 (0.11) | 0.98 | 0.75 |
| 2 | 0.75 (0.30) | 0.77 (0.28) | 1 | 1 |
| 3 | 0.61 (0.36) | 0.62 (0.36) | 0.9 | 0.82 |
| 4 | 0.71 (0.19) | 0.8 (0.19) | 0.79 | 0.63 |
| 5 | 0.77 (0.22) | 0.82 (0.21) | 0.95 | 0.83 |
| 6 | 0.58 (0.27) | 0.64 (0.27) | 0.76 | 0.64 |
| 7 | 0.54 (0.16) | 0.57 (0.14) | 0.57 | 0.47 |
| 8 | 0.98 (0.08) | 0.99 (0.04) | 1 | 1 |
| 9 | 0.57 (0.14) | 0.6 (0.14) | 0.61 | 0.46 |
| 10 | 0.86 (0.18) | 0.96 (0.11) | 0.85 | 0.77 |
| 11 | 0.47 (0.12) | 0.63 (0.14) | 0.55 | 0.46 |
| 12 | 0.93 (0.15) | 0.96 (0.11) | 1 | 1 |
| 13 | 0.63 (0.13) | 0.96 (0.09) | 0.68 | 0.51 |
| 14 | 0.6 (0.24) | 0.86 (0.17) | 0.56 | 0.51 |

[&] Results taken from Toloo and Babaee's [16] study.

The entropies of the Ensemble CCR, Ensemble BCC, Non-Radial and Radial models were 2.616, 2.621, 2.615 and 2.599, respectively. The ME$^{UB}$ from Equation (2i) is 2.639. Comparing the Ensemble CCR with the Non-Radial and Radial CCR models shows that the Ensemble CCR model has a higher entropy. Only the VRS Ensemble BCC model has a higher entropy than the Ensemble CCR model. The standard deviations of the Ensemble BCC model are mostly lower than the CCR model's as well. More importantly, the Ensemble CCR model generates unique rankings for the DMUs, whereas the Non-Radial and Radial models generate a tie for three DMUs. The Ensemble BCC model also generates unique rankings, but the differences occur at the third decimal place. The Ensemble BCC efficiency scores for DMU 10, 12 and 13 were 0.960, 0.959 and 0.962, respectively.

Figures 3 and 4 illustrate the numbers of models (out of 105 total models) where a DMU was fully efficient. These figures are useful for understanding to what extent the assumption $\alpha_1 \approx \alpha_2 \approx \ldots \approx \alpha_n$ was satisfied for the theoretical normal distribution of the ensemble efficiency scores. For these parameters to be similar, the expectation is that a similar number of fully efficient DMUs should exist across all models. Clearly, some DMUs are never fully efficient under any of 105 models and the assumption of identical distributions is violated. While the assumption is violated, Figure 4 illustrates that some DMUs, e.g., 1, 8, 10, 12 and 13, have a somewhat similar number of fully efficient DMUs

to others. These ensemble scores of these DMUs may be considered as normalized random sums generated from identical distributions (such as Distribution 1). All of these DMUs have ensemble efficiency scores greater than 0.95. Similarly, DMUs 5, 6 and 11, in Figure 4, have no fully efficient scores, and these may also be considered as random normalized sums generated from identically distributed pdfs (such as Distribution 2).

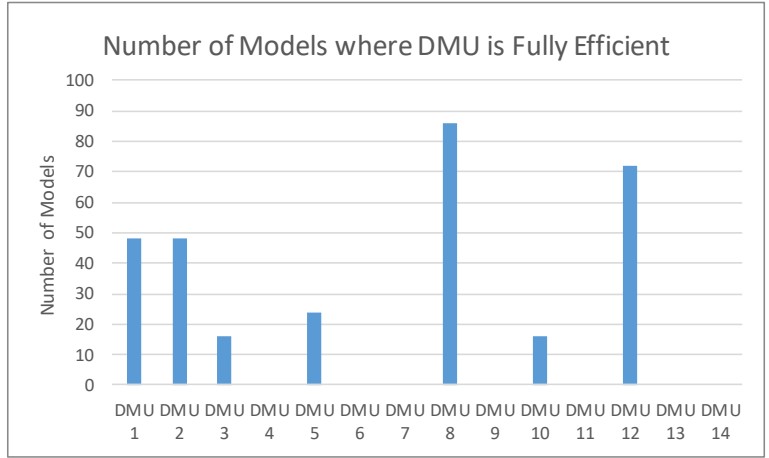

**Figure 3.** Number of times a DMU is fully efficient in Ensemble CCR models.

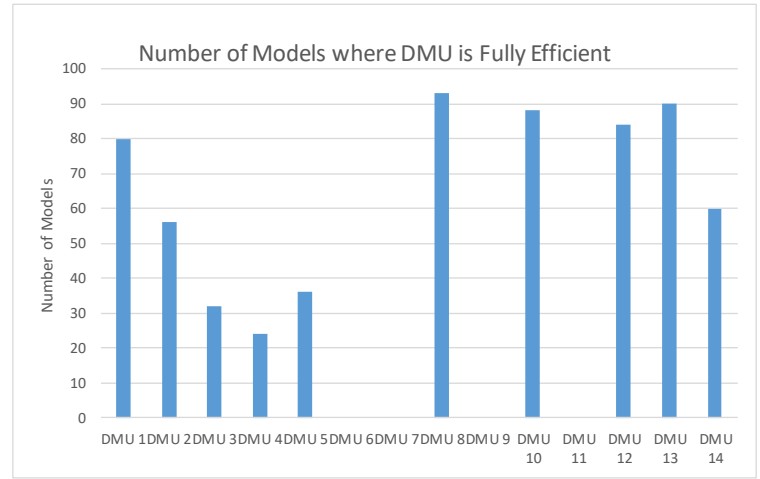

**Figure 4.** Number of times a DMU is fully efficient in Ensemble BCC models.

The ensemble scores for this dataset appear to be random normalized sums from two or more pdfs of the forms given in Equation (2k). Given that these are independent random normalized sums, it can be easily shown that the product of two or more independent MDE pdfs is also an MDE pdf. Figure 5 illustrates two sample MDE pdfs for two different values of alpha. The entropy of an MDE pdf is maximized when the mean of a distribution is greater than 0.5 [27]. For the ensemble BCC model, from Table 3, this criterion is satisfied. The lowest value of the ensemble BCC score is 0.57, which is greater than the mean of 0.5 required to maximize entropy and higher than the lowest values for the efficiency scores for the radial, non-radial and ensemble CCR models. As a result, the ensemble BCC model appears to maximize its entropy slightly better than the ensemble CCR model.

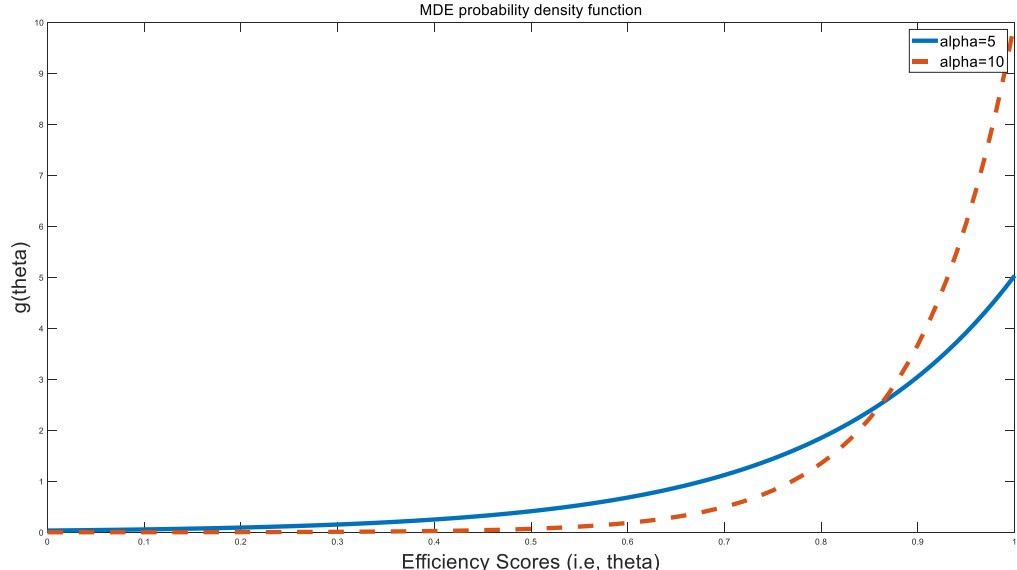

**Figure 5.** The maximum decisional efficiency (MDE) probability density function (pdf) for $\alpha = 5$ and $\alpha = 10$, respectively.

While ensemble scores have a minor violation of an identical distribution for some DMUs, a formal test of the normality of the distribution of the ensemble efficiency scores was conducted. Table 4 illustrates the results of these tests. The Shapiro–Wilk statistic for the Ensemble CCR model is 0.944, and that for the Ensemble BCC model is 0.876, which, at 14 degrees of freedom, are non-significant, consistent the null hypothesis that the efficiency score distribution is normally distributed at the 95% level of statistical significance.

**Table 4.** The results of normality tests.

| | Kolmogorov–Smirnov | | | Shapiro–Wilk | | |
|---|---|---|---|---|---|---|
| | Statistic | *df* | Sig. | Statistic | *df* | Sig. |
| Ensemble BCC | 0.196 | 14 | 0.149 | 0.876 | 14 | 0.051 |
| Ensemble CCR | 0.182 | 14 | 0.200 | 0.944 | 14 | 0.477 |

A paired sample *t*-test for the difference in mean efficiency scores between the Ensemble CCR and the Ensemble BCC models gives a |*t*|-value of 3.524, which is significant at the 99% level of statistical significance (*df* = 13), indicating that a variable returns-to-scale relationship exists between inputs and outputs. The normality of the ensemble efficiency score distributions increases the power of parametric statistical tests.

## 4. Summary, Conclusions and Directions for Future Work

A significant amount of research in the DEA literature has focused on dimensionality reduction/variable selection techniques for small datasets. These techniques are often criticized and have their limitations, with no clear way of selecting which technique is the best. A better approach would be to use an ensemble DEA score that does not make any additional assumptions and provides models that have high entropy values and normally distributed scores under restrictive assumptions. Pendharkar [14], in his study, has already provided a theoretical foundation for the reliability of ensemble DEA scores. The added benefit of ensemble DEA scores is that they provide unique DMU rankings.

The normality of ensemble DEA scores is not guaranteed unless the ensemble DEA scores are normalized sums generated from independent identically distributed MDE pdfs. This assumption

may not be strictly satisfied in most real-world datasets, but the current study shows that minor deviation from this assumption may be tolerated because the entropy of all MDE pdfs is maximized when normalized sums have a value greater than 0.5. This means that, typically, the differences in means between the underlying pdfs (Equation (2l)) for ensemble entropy scores will be less than 0.5, and, while these pdfs may not be identically distributed, the means of these distributions will be close, resulting in the likely normal distribution of ensemble scores in most real-world cases. The normality of ensemble DEA scores allows for the application of traditional statistical tests for return-of-scales hypothesis tests. Traditional DEA hypothesis-testing methods are not perfect and are known to be slightly biased [28]. Future research may focus on comparing ensemble DEA-based hypothesis testing with traditional DEA hypothesis testing to identify which method provides reliable results.

**Funding:** This research received no external funding.

**Conflicts of Interest:** The authors declare no conflict of interest.

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
