# Peer review of "A Comparison of Ensemble and Dimensionality Reduction DEA Models Based on Entropy Criterion"

_algorithms, doi:10.3390/a13090232_

Round 1

Reviewer 1 Report

  1. The author should show the descriptive statistics of the data used in this study and give some explanation on the trends in the input and output factors during the study period at least.
  2. The author should show the list of DMUs.

Author Response

Response 1.

Table 3 was redrawn to report standard deviations of scores next to means.  Figures 3 and 4 were added to illustrate number of fully efficient DMUs for different models. We did not collect the dataset so trends information was not available to us.

Response 2.

List of DMUs are now shown in Table 3 and Figures 3 and 4.

Reviewer 2 Report

  1. The abstract should be improved including one sentence explaining what is the knowledge gap of research and more general conclusions.

  1. The introduction misses significant information, such as: What is the research question? What makes the applied methodology suitable and superior in comparison to existing studies?, and What is the expected new insight gained by applying the methodology?.

  1. Methods must include tests used to guarantee reliability and robustness of results.

  1. Discussion should be improved including strengths and shortcomings of this analysis, the meaning of results in comparison with literature findings and a short outlook on further research requirements and possible research extensions.

I wish that these comments can help the author to improve the paper.

Author Response

  1. Entire abstract was rewritten.
  2. The third paragraph was modified and rewritten to highlight contribution of the study.

  3.  Sections 2 and 3 were entirely rewritten and Figures were added.  Table 4 was also added to include statistical tests.

  4. Last Section 4 was expanded to highlight short comings of the study.

Reviewer 3 Report

The paper "A Comparison of Ensemble and Dimensionality Reduction DEA Models Based on Entropy Criterion" studies some properties of ensemble DEA models (proposed earlier), and claims that the proposed method produces scores for DEA models which can help in creating a unique ranking of DEA models and the sores have high entropy and are normally distributed. The paper is general is well written and easy to follow. However, I have some fundamental questions for the conclusions made:
There is no theoretical base provided to justify the claims made by the author. The conclusions are drawn by testing the model on a single data set. There is no guarantee that the scores will be normally distributed if the data set is changed. Also, the claim that the entropy is high and is very close to the maximum model entropy is not backup up by sufficient evidence. This is tested on a single small data set. The explanation: "Part of the reason for the stability of ensemble DEA efficiency scores
160 is that, even for small datasets, some DEA models in an ensemble will always satisfy the heuristic
161 given in eq. (1.1), which will increase the reliability of the ensemble efficiency scores." is weak and not sufficient to justify the claims made by the author.
Hence, the paper needs a major revision to address these fundamental concerns and state the exact contributions accordingly.

Author Response

Theoretical foundation for statistical distribution was provided at the end of Section 2.  The highlighted text section was modified.

Reviewer 4 Report

Major   - The major concern is about the properties of the ensemble DEA model (maximisation of the entropy and normal distribution), which are not formally proven, but only verified at empirical level with some applications.   Minor   -  All the equations in the manuscript are numbered, but some of them are not recalled in the text. So, some equation labels can be dropped. Moreover, it is better to align the labels assigned to the constraints of the same model.   - The caption of Table 1 is in a separate page with respect to the relevant table.   - What does VRS stand for?

Author Response

Theory for normality of distribution was provided at the end of Section 2.  The type of distributions (MDE) maximize entropy at the value of 1 (Figure 5).  The equation numbers were removed for uncalled equations.  VRS was mentioned. 

Round 2

Reviewer 2 Report

Authors shows changes in the paper, according to my recommendations.

I think that paper is adequate to publish.

Author Response

thanks

Reviewer 4 Report

The author has addressed all the critical point rises in my report and I found that the quality of the paper is now improved. I have only a minor comment about the fact that the Central Limit Theorem provides an asymptotic and not an exact result for the normality of the scores. This should be specified in the manuscript.

Author Response

The asymptotic convergence was mentioned in section 2.